# Fabrication of Anti-Reflective Surface with Superhydrophobicity/High Oleophobicity and Enhanced Mechanical Durability via Nanosecond Laser Surface Texturing

**DOI:** 10.3390/ma13245691

**Published:** 2020-12-13

**Authors:** Huixin Wang, Jian Zhuang, Jiangtao Yu, Hongyan Qi, Yunhai Ma, Hubiao Wang, Zhijun Guo

**Affiliations:** 1College of Biological and Agricultural Engineering, Jilin University, Changchun 130022, China; wanghx17@mails.jlu.edu.cn (H.W.); zhuangjian_2001@163.com (J.Z.); yujt18@mails.jlu.edu.cn (J.Y.); qihy725@163.com (H.Q.); hb_wang@jlu.edu.cn (H.W.); 2Key Laboratory of Bionic Engineering, Jilin University, Changchun 130022, China; 3College of Vehicle and Transportation Engineering, Henan University of Science and Technology, 48 Xiyuan Road, Jianxi District, Luoyang 471003, China; gzhj1970@163.com

**Keywords:** laser surface texturing, chemical treatment, anti-reflective, superhydrophobic, mechanical durability

## Abstract

In this work, anti-reflective surface with superhydrophobicity/oleophobicity and enhanced abrasion resistance was fabricated on steel alloy surface. Two different surface patterns (i.e., parallel microgrooves and spot arrays) were created by nanosecond laser ablation and chemical immersion. The surface micro/nanostructure, spectral reflectance, wettability, and abrasion resistance of all the samples were determined. The experimental results showed that the laser-chemical treated surfaces exhibited much lower spectral reflectance and significantly enhanced surface integrities compared with the untreated surface. Firstly, the contact angles of water, glycerol, and engine oil on the laser-chemical treated surfaces were increased up to 158.9°, 157.2°, and 130.0° respectively, meaning the laser-chemical treated surfaces achieved both superhydrophobicity and high oleophobicity. Secondly, the laser-chemical treated surface showed enhanced abrasion resistance. The experimental results indicated that the spectral reflectance of the laser-chemical treated surfaces remained almost unchanged, while the laser-chemical treated surface patterned with parallel microgrooves sustained superhydrophobicity with a water contact angle of 150.2° even after more than one hundred abrasion cycles, demonstrating the superior mechanical durability. Overall, this fabrication method has shown its effectiveness for fabrication of multifunctional metal surface integrating the surface functionalities of anti-reflectivity, superhydrophobicity/high oleophobicity, and enhanced abrasion resistance.

## 1. Introduction

The use and cost of energy significantly affect our daily lives. The World Energy Council predicts that by 2050 primary energy demand will triple because of the growing population and the improving living standards of developing countries [1]. An energy crisis is emerging as the conventional energy resources, such as natural gas, oil, or coal, are rapidly running out. Therefore, searching for new energy resources has been a critical issue for the sustainable development of human beings. 

Among all the alternative energy resources, solar energy has attracted the attention of researchers due to its nature of cleanliness that can be supplied without any environmental pollution [2]. One of the main components of the solar energy system is solar absorber, which is a solar collection device that can transfer solar radiation energy to heat first and other energies such as electrical energy and thermal energy subsequently [3]. In recent decades, anti-reflective surface has been widely utilized in the solar absorber to improve the efficiency of energy transfer. Meanwhile, anti-reflective surfaces have also been a key component in many optoelectronic industries, such as sensing, photovoltaics, and photocatalysis [4]. There have been a lot of technologies used to fabricate the anti-reflective surface, such as surface coating and patterning [5,6]. Although there have been a lot of research works that successfully achieved highly efficient light absorption capability, there are a few distinct drawbacks of these surfaces, such as poor waterproofness and low mechanical durability, which has significantly prevented them from being adopted for industrial applications. 

To solve the problem, modification and enhancement of surface integrity and reliability of anti-reflective surfaces have always been of vital importance for the scientific community. The first key surface integrity that requires enhancement is surface wettability, which aims to render the surface superhydrophobicity/oleophobicity [7]. Superhydrophobic surfaces exhibit a number of functionalities such as anti-icing, self-cleaning, anti-fouling, and anti-adhesion, which ensures they can be applied in various types of extreme environments (e.g., weathers with acid rain and tendency of ice formation) [8,9,10,11]. Basically, fabricating an anti-reflective surface while maintaining superhydrophobicity/high oleophobicity will enable many commercial applications such as metal-based solar absorber [12], solar heater [13,14] or thermoelectric generator [15,16]. Thus, it is highly essential to fabricate an anti-reflective surface with enhanced surface wettability [17,18]. The second key surface integrity that requires enhancement is mechanical durability. On many occasions, commercially available anti-reflective and superhydrophobic surfaces have to be applied under severe abrasion conditions, such as the desert where sand blowing occurs frequently. Generally, the wettability of superhydrophobic surfaces will be significantly altered after being abraded by another surface for a certain time duration [19,20], meaning that they will be no longer superhydrophobic. Therefore, many researchers have attempted to enhance the abrasion resistance of superhydrophobic surfaces using various fabrication methods. For example, Liu et al. fabricated a superhydrophobic surface on X80 pipeline steel substrates and the wettability of the surface was sustained after 10 cycles of abrasion [21]. Zhu et al. created a superhydrophobic surface by a hot-pressing approach followed by Ag deposition and surface fluorination, and the experimental results showed that high water contact angle and intact surface textures were well preserved after 10 abrasion cycles [22]. However, these studies showed that the superhydrophobic surfaces could only retain mechanical durability with a limited number of abrasion cycles and low contact pressure. Therefore, it is important to fabricate superhydrophobic and anti-reflective surfaces with enhanced mechanical durability that can sustain more abrasion cycles for practical applications. 

For practical applications, processing efficiency and production cost are two major factors that should be taken into consideration. Traditional coating or nanofabrication methods used to produce the multi-functional surface usually take a while (3–5 h). Compared with these traditional fabrication techniques, laser-based texturing has demonstrated its effectiveness for modifying the optical properties and wettability of an engineering alloy surface due to the advantages of high processing efficiency, process flexibility, ease of automation, reduced production cost, and environmental friendliness [23,24,25,26,27]. For example, Vorobyev et al. fabricated a multifunctional surface exhibiting combined effects of enhanced light absorption and superhydrophobicity by femtosecond laser texturing [28]. However, the existing laser-based fabrication techniques rely on the generation of a periodic micro-/nano-scale surface structure with very fine spatial resolution, resulting in relatively low process throughput, which will make it difficult to scale up the process [29,30,31]. The high capital cost associated with the ultrashort pulsed lasers also makes the process very expensive [28]. Therefore, development of a more time-efficient and cost-effective laser-based process for fabrication of an anti-reflective and superhydrophobic surface is essentially needed by the laser texturing community. 

In our previous study, we developed a time-efficient and cost-effective laser-based surface texturing technique, and fabricated a functional surface that not only showed superhydrophobicity but also exhibited improved corrosion resistance in both marine atmospheric and underwater environments [32]. In this work, more surface functionalities such as spectral reflectance, surface wettability, and mechanical durability of the laser-chemical treated surfaces were systematically characterized and investigated. Through surface characterizations, the surface treated using this technique exhibits combined functionalities of low spectral reflectance, superhydrophobicity/high oleophobicity, and enhanced abrasion resistance. This work provides detailed analyses and insights for how laser processing parameters and chemical treatment affect these functionalities. Compared with other fabrication processes, such as coating [33,34,35,36,37], nanofabrication [38] and ultrafast laser irradiation [39], the laser-chemical treatment method developed in this work exhibits advantages in capability to achieve multiple surface functionalities, high processing efficiency and low production cost. This work can be a very effective path for fabrication of an anti-reflective surface with enhanced surface wettability and mechanical durability, which will provide good potential for improving the surface integrity and reliability of optical devices such as thermoelectric generators, electro-optical cells and solar hot water tanks.

## 2. Materials and Methods 

### 2.1. Laser-Chemical Treatment

A total of 1095 alloy steel (AISI 1095) sheets with a dimension of 25.4 × 25.4 × 0.635 mm^3^ were used in this research work, which has superior mechanical, thermal and tribological properties and has been widely used in many industrial applications [40,41,42]. Laser surface texturing experiments employed a laser marking machine (Han’s Laser, Shenzhen, China) with a center wavelength of 1064 nm, as shown in Figure 1a. In this work, the laser texturing experiment utilized the following processing parameters: laser beam diameter of 200 µm, repetition rate of 5000 Hz, pulse energy of 2 mJ, laser fluence of 6.37 J/cm^2^, pulse duration of 10 ns, power intensity of 0.32 GW/cm^2^ and average power of 10 W. The focal length was 266 mm and offset distance between focal plane and metal surface was 10 mm. Two periodic surface patterns (i.e., parallel microgrooves (Pattern I) and spot arrays (Pattern II)) with a laser process area of 20 × 20 mm^2^ were created along the y-axis on the laser textured surface, which can be found in Figure 1b,c. The pitch between each spot or microgroove (X) was 800 µm for both patterns, and the distance between two laser pulses (Y) was 100 and 800 µm for pattern I and pattern II, respectively. The scanning speed was 500 and 4000 mm/s for pattern I and pattern II, respectively, while the process rate can be up to 238 and 2000 cm^2^/min, respectively, which ensures high-speed processing of the AISI 1095 samples. With the parameters of spacing and process rate, the density for pattern I and pattern II are 12.5 lines per cm and 156 spots per cm^2^, respectively. The pulse per irradiation point was 1 and 2 for pattern I and pattern II, respectively, while the overlapping ratio for pattern II was 39%. The laser textured samples were subsequently immersed in an ethanol solution (Purity: 99.5%), which consists of 1.5 wt.% 1H,1H,2H,2H-perfluorooctyl (FOTS) silane reagent for ~1 h. The samples were rinsed by Deionized water and then kept at 80 °C in a vacuum oven for 20 min until they were completely dried out. Sample type, surface pattern and chemical treatment used for different samples fabricated in this work can be found in Table 1. Sample A is the sample without any treatment; Sample B is the smooth sample that is chemically treated only; Samples C and D are the laser textured samples using patterns I and II respectively, and both of them have been chemically treated; and Sample E is the laser textured sample using pattern II without the chemical immersion treatment.

### 2.2. Surface Characterizations

Surface morphology and roughness was tested by a non-contact 3-D laser scanning confocal microscope (OLS5000, Olympus, Tokyo, Japan). According to the international standard (ISO 25178) [43], the profile roughness was characterized at five different locations, and then the average values were calculated as the final results. The test direction was vertical to the groove/spot pattern. The surface structure was examined by a scanning electron microscope (SEM, Evo18, Zeiss, Oberkochen, Germany), where the SEM images were taken at acceleration voltages of 1.8–2.0 kV. Surface spectral reflectance measurements were performed for the laser-chemical treated samples using a UV-VIS-NIR spectrometer (UV2600, Shimadzu Corporation, Kyoto, Japan) with normal incidence. The spectrometer characterizes the spectral reflectance of the sample surface within wavelength regions ranging from 450 to 1670 nm. According to the International Commission on Illumination (CIE) classification, the wavelength range used in this work can be divided into a visible region from 450 to 780 nm and an IR region from 780 to 1650 nm. Calibration was performed before the measurement by measuring the incident flux remaining in the integrating sphere after reflecting from standard reference material. The measurement results were analyzed to process and visualize the actual spectral reflectance. For each sample, five measurements were taken at different positions and the average value was calculated and reported.

Contact angle measurements were conducted to examine the wettability of the samples using a contact angle goniometer (JC2000A Powereach, Powereach Company, Shanghai, China). Three different liquids including DI water, glycerol (Sigma-Aldrich, St. Louis, MO, USA), and engine oil (AeroShell Piston Engine Oil 100, AeroShell, The Hague, The Netherlands) were employed for wettability testing, and their physical properties are listed in Table 2. Since the contact angle of the laser textured surface evolves with time, the measurements were conducted within one hour after the fabrication of the samples. For each measurement, a liquid volume of 4 µL was dropped onto the sample surface forming a still droplet, and its optical shadowgraph was obtained and then quantitatively analyzed using the ImageJ software (ImageJ 1.50b) to determine the static contact angle for each measurement. The contact angle hysteresis was measured using the titling base unit accessory. After the droplet was applied on the flat surface, the stage was titled for 1.0° per time until the droplet started to move. Contact angle hysteresis was defined as the difference between the downside contact angle (advancing angle) and upside contact angle (receding angle) of the droplets when the droplet starts to move. Six measurements were performed at various locations for each sample surface, and the average value of the measurement results was reported. 

Scratch test was designed to evaluate the mechanical durability of the laser-chemical treated samples under severe abrasion conditions. During the scratch test, an 800 grit SiC sandpaper was selected as the abrasive surface. The laser-chemical treated surfaces were tested facing the abrasive surface with a fixed load of 100 g. The tape was sticked between the weight and the back of the sample with a linear speed of around 1 mm/s, which the sample and the weight can move together along with the dragging of the tape. One scratch cycle was defined as a relative reciprocating movement distance of 100 mm (along the X-axis in Figure 1), as shown in Figure 2. The spectral reflectance of the samples was measured after 120 scratch cycles (completion of the scratch test), while the contact angles of the samples were measured at five different locations after every 10 cycles and the average value was reported. Energy dispersive X-ray spectroscopy was carried out for the laser-chemical treated samples before and after the scratch test in order to evaluate how scratch test will affect the surface chemical composition. 

## 3. Results

### 3.1. Surface Morphology and Microstructure 

3-D surface profiles of the laser-chemical treated surfaces (samples C and D) reveals how laser texturing affects surface morphology as shown in Figure 3a–c. The 3-D surface profiles of the laser-chemical treated surfaces clearly demonstrate that the surface pattern was created based on the laser scanning strategy, while the periodic surface structure was generated as a result of material removal induced by laser ablation. It is clearly shown in Figure 3a–c that laser surface texturing creates the surface structure with an etch depth of 150 µm, a feature size of ~200 µm and a pitch of ~800 µm. At the same time, S_a_ (defined as the arithmetic mean height of a surface) and S_z_ (defined as the maximum height of a surface) were measured for all the samples and the measurement results can be found in Figure 3d. It is shown that samples A and B exhibit very low surface roughness values of S_a_ (2.11 ± 1.05 and 2.55 ± 1.03 µm) and S_z_ (2.42 ± 1.09 and 4.98 ± 1.20 µm), indicating that the chemical treatment has trivial impact on surface roughness, and samples C, D and E exhibit S_a_ values of 32.71 ± 3.25, 15.25 ± 2.54 and 30.63 ± 3.10 µm, and S_z_ values of 152.3 ± 5.25, 150.48 ± 6.25 and 151.23 ± 4.58 µm, respectively, which are relatively higher than those of the untreated surface and chemical treated surface. These experimental results indicate that laser surface texturing induces distinct increase of surface roughness, while the surface roughness is also demonstrated to be highly dependent on surface patterns. The change of surface roughness is considered as a key factor that affects the surface functionalities (e.g., wettability and reflectance of the laser textured surface) [5,44].

The SEM images of the untreated sample and laser-chemical treated samples are shown in Figure 4. It can be observed in Figure 4a that the untreated surface (sample A) showed an isotropic texture with a relative smooth surface. However, from the SEM images of laser-chemical treated samples, sample C exhibited periodic dual-scale micro/nanostructures consisting of nanoparticles covered on top of the microgrooves (Figure 4b,d) under both low and high magnifications (50× and 20,000×). While sample D was filled with periodically arrayed microholes with some nano-scale features on the hole periphery as well (Figure 4c,e). It is clearly found from Figure 4b that the laser pulses were overlapped when treating sample C. Due to the overlap of laser pulses, the number of laser pulses per irradiation point became higher, which leads to repeated heating and cooling in a unit area. This significantly increases the density of micro/nanostructures on sample C, resulting in more surface roughness features compared with that of sample D as shown in Figure 4c. 

### 3.2. Spectral Reflectance

Spectral reflectance measurement results for all the samples are shown in Figure 5. The refractive indices of 1095 steel and air is about 128 and 1, respectively. The spectral reflectance of all the samples shows a rising tendency with the increase of wavelength. The spectral reflectance for sample A falls 42%–46% within the visible spectrum and 46%–71% within the IR spectrum. After chemical treatment, the spectral reflectance for sample B was decreased to 17%–18% within the visible spectrum and 18%–34% within the IR spectrum. After laser texturing process, the spectral reflectance of sample E was 7%–9% within the visible spectrum and 9%–21% within the IR spectrum, which was much lower than the spectral reflectance of sample B, indicating that laser texturing process have more pronounced effect for decreasing the spectrum reflectance compared with that of chemical immersion process. Meanwhile, after laser-chemical treatment, the spectral reflectance for samples C and D was dramatically reduced to 4%–6% within the visible spectrum and 6%–20% within the IR spectrum. This indicates the spectral reflectance of laser-chemical treated surfaces drops significantly over the entire measured wavelength range compared to that of the untreated surface, chemically treated surface and laser textured surface. In the meantime, it should be noticed that the surface color of the laser-chemical treated samples turned black, which could be attributed to the oxidation during the laser texturing process and the deposition of FOTS coating layer onto the surface, indicating the developed process can be used to colorize metal surface. 

The wide-band anti-reflective properties of the laser-chemical treated surfaces demonstrate the strong capability of the developed process for altering the optical properties of metal surface. The distinct anti-reflective property should be attributed to both the surface micro/nanostructures and the modification of surface chemistry on the laser-chemical treated surfaces, which leads to the diffuse reflection and multiple specular reflection of light [45,46]. Firstly, the nanostructure with feature size smaller than light wavelength can reduce the reflectance owing to the graded refractive index formed by subwavelength surface textures at the air-solid interface [47]. Secondly, the microstructure with feature size greater than light wavelength can enhance the anti-reflectance effect by trapping light in surface cavities and the Fresnel angular dependent reflection [28]. Thirdly, it is known that the silane reagent has a low reflective index of 1.3 at room temperature. Therefore with the formation of an anti-reflective silane nanostructure layer due to the chemical immersion treatment, the spectral reflectance is further reduced [48]. All of the above-mentioned mechanisms lead to the low spectral reflectance of the laser-chemical treated surfaces, which could be attributed to the modulation effect of optical property induced by the combination of laser texturing and chemical immersion processes. It is also observed that samples C and D exhibited quite identical anti-reflective capabilities in the visible and IR spectrum. This could be attributed to the similar feature size (hole diameter and depth/groove width and depth) fabricated on the laser-chemical surfaces using the same laser power intensity, which renders them similar light-trapping capabilities [49].

### 3.3. Surface Wettability

The wettability of samples A–D in terms of water, glycerol, and engine oil were experimentally evaluated through contact angle measurements. From Figure 6a,b, it can be found that the untreated surface (sample A) is hydrophilic with a water contact angle of 81.9° ± 2.5°. After chemical treatment, the water contact angle of sample B was increased to 113.8° ± 2.0°, indicating that the surface became hydrophobic by chemical treatment only. After laser texturing, sample E exhibits superhydrophilicity with the water contact angel of 8.3° ± 2.3°. After laser-chemical treatment, the water contact angle was increased to 158.9° ± 2.1° and 153.2° ± 1.6° for samples C and D, respectively. At the same time, the contact angle hysteresis for samples C and D were 4.2° ± 1.3° and 5.9° ± 1.3°, respectively. All of these indicates that both of the surfaces achieved superhydrophobicity after laser-chemical treatment, which clearly demonstrates that surface chemistry and surface structure of laser-induced micro/nanostructured surfaces are equally important to achieve the target wettability condition, and thus the final wetting performance of the surface is a complex combination of surface micro/nanostructures and surface chemistry. 

Besides water, the contact angles of glycerol and engine oil on all the samples were also characterized in Figure 6 in order to demonstrate their wetting properties in terms of organic liquids. For glycerol, the contact angles of both sample C and sample D were 157.2° ± 1.8° and 152.7° ± 2.3°, while the contact angles of samples A, B and E were 79.9° ± 2.3°, 100.2° ±1.5° and 9.5° ± 1.5°, respectively. For engine oil, the contact angles were 130.0° ± 1.6° and 123.0° ± 2.1° for sample C and sample D, while the contact angles were only 62.3° ± 1.6°, 88.2° ± 1.5° and 9.0° ± 1.7° for samples A, B and E, respectively. This clearly indicates that the laser-chemical treatment method can induce high oleophobicity or even superoleophobicity displaying high contact angle in terms of organic liquids. Compared with the water contact angle, the contact angles of glycerol and engine oil on all the samples became smaller. This is mainly attributed to the difference in surface tension between water and glycerol/engine oil. Based on literature data, the surface tension of water is 72.8 mN/m while the surface tensions of glycerol and engine oil are 64 and 30.3 mN/m, respectively [50,51,52]. The liquid with lower surface tension tends to spread across the surface more easily and rapidly, leading to a smaller contact angle [50]. 

### 3.4. Mechanical Durability

Mechanical durability of anti-reflective surfaces is the primary concern that prevents them from being adopted for commercial applications. In this work, the mechanical durability of the laser-chemical treated anti-reflective and superhydrophobic surfaces was evaluated by scratch test [53,54]. Spectral reflectance of all the samples was measured before and after 120 scratch cycles, as shown in Figure 7a. For the untreated surface, the spectral reflectance was reduced after the scratch test. This should be attributed to the fact the surface roughness was increased during the scratch test, while the higher surface roughness will result in the lower spectral reflectance. However, the spectral reflectance of sample B was increased after the scratch test, as the silane nanostructure layer with the low reflective index was damaged after the scratch test. For the laser textured surface, the spectral reflectance of sample E remained almost unchanged during 120 scratch cycles, since the surface nearly retained its original micro/nanostructures and there was no silane layer on the surface. For the laser-chemical treated surfaces, samples C sustained its low spectral reflectance after the scratch test, while the spectral reflectance of sample D was increased slightly, which indicates that their optical property remained relatively stable during the scratch test. During the scratch test, the micro/nanostructures formed on the laser-chemical treated surfaces was well preserved, thus the spectral reflectance of the surfaces remained nearly identical before and after the scratch test. However, due to the lower surface roughness, the silane layer on the surface of sample D was destroyed more severely, resulting in the slight increase of spectral reflectance.

At the same time, water contact angles were recorded at every 10 scratch cycles until the completion of 120 cycles as shown in Figure 7b,c. For sample A, as the number of scratch cycles increased, the water contact angle was continually reduced and reached 32.4° after 120 cycles. At the same time, after 20 scratch cycles, sample B lost its hydrophobicity and the surface became hydrophilic with the water contact angle of 86.0°. Then, as the number of scratch cycles increased to 80, the surface of sample B was completely destroyed while the values of contact became identical to those of sample A eventually. As for the laser-chemical treated surfaces, sample C showed excellent mechanical durability with a very minimal decrease of water contact during the scratch test. Its superhydrophobicity was retained after 120 cycles with a water contact angle of 150.2°. However, the contact angle of sample D drastically decreased to 90.0° after 120 scratch cycles, indicating the surface has lost its superhydrophobicity and turned hydrophobic. For the laser textured surface, sample E kept its low water contact angle during the scratch test as strong abrasion has very minimal effect on the superhydrophilicity of a surface. All of these demonstrate that the laser-chemical treated sample with the parallel microgroove pattern shows excellent abrasion resistance. 

Table 3 shows the chemical composition of samples A–E before and after scratch test. After 120 scratch cycles, the weight percentage of elements fluorine (F) and silicon (Si) for sample B was dramatically reduced, illustrating that the FOTS layer was partially damaged during the scratch test. Meanwhile, the weight percentage of element F for samples C and D was reduced by 6.68% and 15.21%, respectively. The lower weight percentage decrease of element F for sample C indicates that the FOTS layer on sample C was more stable during the scratch test, which corresponds well with its high water contact angle and low spectral reflectance sustained during the scratch test. 

The experimental results clearly illustrate that surface pattern plays a key role in improving the abrasion resistance of the laser-chemical treated surface. Figure 8 showed the SEM images of laser-chemical treated surfaces before and after 120 scratch cycles. Compared with the surface before scratch test in Figure 8a, some scratches can be observed on the surfaces of sample C and sample D after 120 scratch cycles, as shown in Figure 8b,c. For sample C, fewer scratches can be observed after abrasion, which suggested that the chemical layer on sample C was slightly damaged during abrasion. However, there are a lot more scratches on the surface of sample D after abrasion, which indicates that sample D has been more severely destroyed, resulting in the dramatic decrease of water contact angle value. This clearly indicates that sample C exhibited excellent mechanical durability while sample D exhibited distinctly worse abrasion resistance, which could be potentially explained using a microstructural point of view. According to the SEM images as shown in Figure 4b,c, the parallel microgroove patterns exhibit continuous surface structure while the holes inside the spot array patterns are isolated. Bump features were clearly formulated on the laser textured areas of sample C as a result of cyclic heating and cooling in the localized area during the interaction between the laser beam and the substrate material. Therefore, the peak to valley height on sample C is much larger than that on sample D. The continuous surface micro/nanostructure with a larger peak to valley height are more likely to be better preserved during the cyclic abrasion with the SiC sandpaper [55,56]. With less damaged surface structure, sample C sustained continued superhydrophobicity, and thus exhibited much better abrasion resistance. Besides, the higher density of dual-scale micro/nanostructures in sample C also contributes a lot to the enhancement of abrasion resistance [57]. This laser-chemical treated technique demonstrated promising effectiveness in enhancing the robustness and reliability of metal alloy and could have more practical applications that require the production of the anti-reflective surface with superhydrophobicity/oleophobicity and good mechanical durability.

## 4. Conclusions

In this work, an anti-reflective surface with superhydrophobicity/oleophobicity and enhanced abrasion resistance was fabricated using a laser-based surface texturing technique that combines nanosecond laser ablation with chemical immersion treatment. Two different periodic surface patterns (i.e., parallel microgrooves and spot arrays) were fabricated and dual-scale surface structures were observed on the surfaces. This laser-chemical treatment method renders some outstanding functionalities for the steel alloy surface:The spectral reflectance of the laser-chemical treated surface was significantly reduced compared with that of the untreated surface.The laser-chemical treated surfaces showed strong repellency to both water and organic liquids, meaning the surfaces achieved both superhydrophobicity and high oleophobicity.The abrasion resistance of the laser-chemical treated surface has been significantly improved, while both the contact angle and spectral reflectance for the laser-chemical treated surface with microgroove pattern remained almost unchanged after 120 abrasion cycles.

Therefore, this laser-chemical treatment method has been demonstrated as an efficient method for the fabrication of the anti-reflective metal surface that integrates other outstanding properties of superhydrophobicity/high oleophobicity and enhanced mechanical durability. This method could lead to more applications in diverse areas including optical, sensor, transportation, and energy industries. 

## Figures and Tables

**Figure 1 materials-13-05691-f001:**
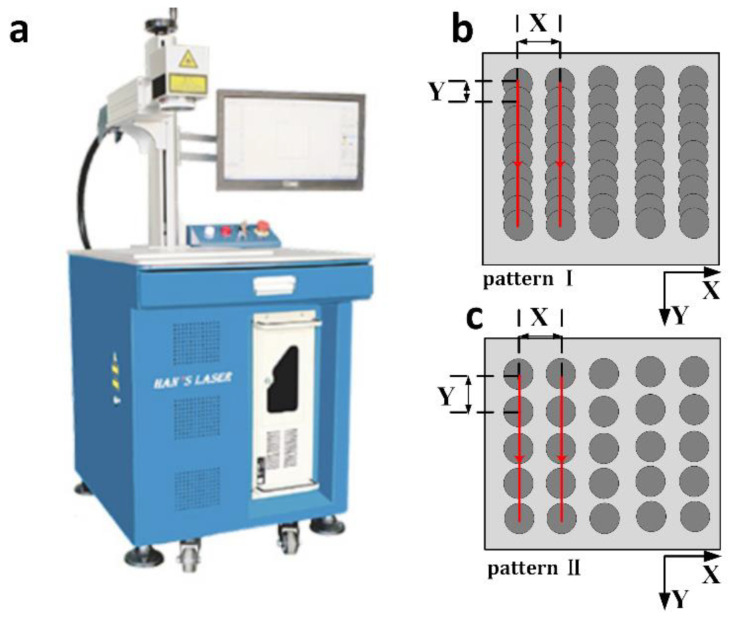
(**a**) Han’s Laser marking machine; and pattern design schematics of (**b**) parallel microgrooves (pattern I) and (**c**) spot arrays (pattern II) fabricated by laser surface texturing.

**Figure 2 materials-13-05691-f002:**
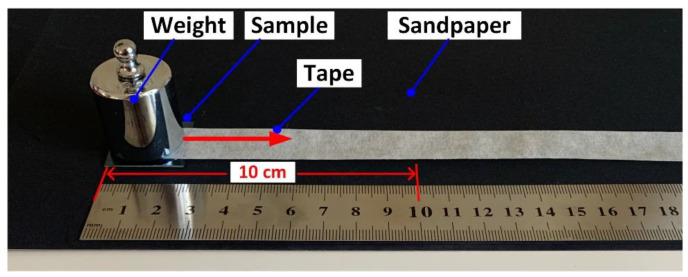
Schematic illustration of the experimental setup for the scratch test in this research work.

**Figure 3 materials-13-05691-f003:**
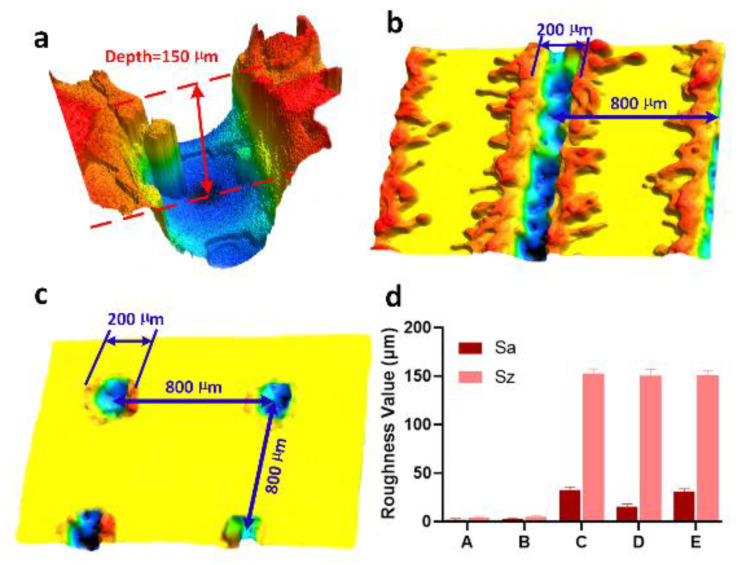
Surface morphology and roughness of different samples: (**a**) 3-D profile of a hole created by single laser pulse; (**b**) 3-D surface profile of sample C; (**c**) 3-D surface profile of sample D; (**d**) S_a_ and S_z_ measurement results of for samples A–E.

**Figure 4 materials-13-05691-f004:**
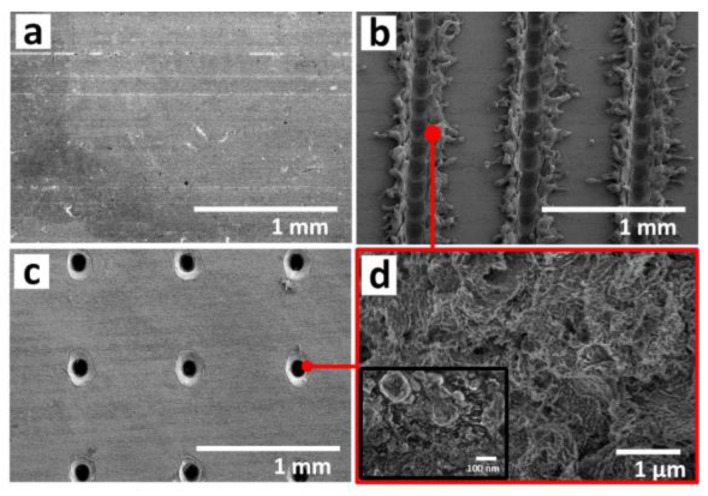
SEM images of different samples: (**a**) Low magnification SEM images for sample A (50×); (**b**) low magnification SEM image for sample C (50×); (**c**) low magnification SEM image for sample D; (**d**) zoom-in view of sample C and D (20,000×, 50,000×) shows the dual-scale micro/nanostructures induced by laser chemical treatment [32].

**Figure 5 materials-13-05691-f005:**
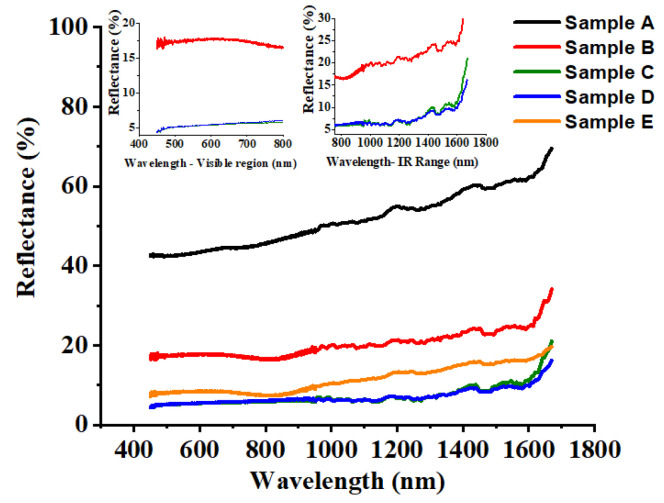
Spectral reflectance measurement results for samples A–E. In this work, the wavelength range for the reflectance measurement is from 450 to 1650 nm. According to the international standards, the wavelength ranging from 450 to 780 nm belongs to the visible spectrum and the wavelength ranging from 780 to 1650 nm belongs to the IR spectrum.

**Figure 6 materials-13-05691-f006:**
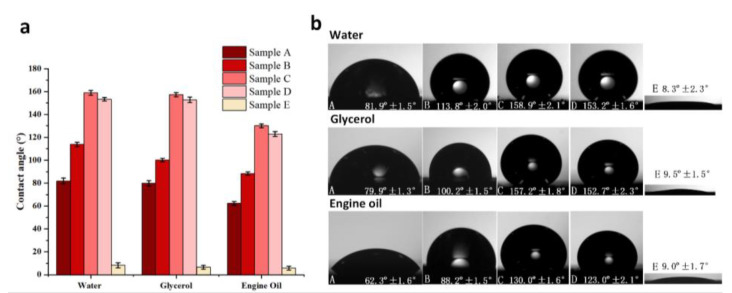
(**a**) Contact angle measurement results of water, glycerol and engine oil for samples A–E; and (**b**) optical shadowgraphs of the static water droplets on these samples during the measurements.

**Figure 7 materials-13-05691-f007:**
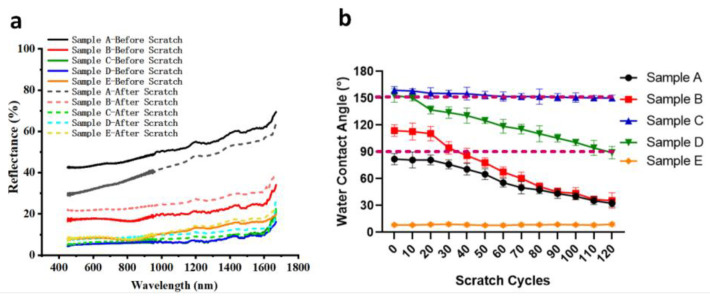
(**a**) The spectral reflectance of samples A–E before and after 120 scratch cycles; (**b**) the variation of water contact angle on samples A–E as a function of scratch cycles during the scratch tests.

**Figure 8 materials-13-05691-f008:**
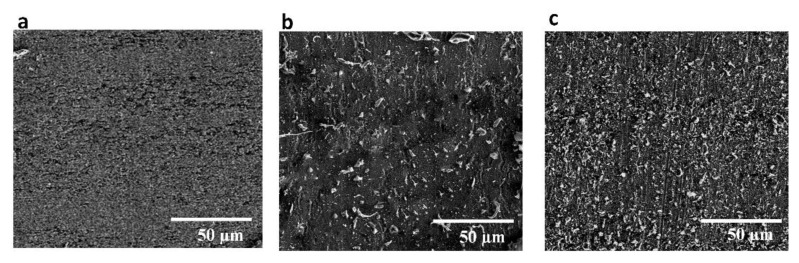
(**a**) SEM image (1000×) for laser-chemical treated surfaces before scratch test. SEM images (1000×) after 120 scratch cycles for (**b**) sample C and (**c**) sample D.

**Table 1 materials-13-05691-t001:** Sample type, surface pattern, and chemical treatment reagent used for different samples fabricated in this work.

Sample Type	Surface Pattern	Chemical Treatment Reagent
A	/	/
B	/	FOTS
C	pattern I	FOTS
D	pattern II	FOTS
E	pattern II	/

**Table 2 materials-13-05691-t002:** Physical properties of the different liquids used in this work.

	Water	Glycerol	Engine Oil
Density	1 g/cm^3^	1.261 g/cm^3^	0.887 g/cm^3^
Melting point	0 °C	18 °C	−18 °C
Boiling point	100 °C	290 °C	780 °C
Viscosity (under 25 °C)	0.89 mPa·s	600 mPa·s	180 mPa·s
Surface tension	72.8 mN/m	64 mN/m	30.3 mN/m

**Table 3 materials-13-05691-t003:** Chemical composition of samples A–E before and after scratch test.

	Before Scratch Test					After 120 Scratch Cycles				
	Fe	C	O	F	Si	Fe	C	O	F	Si
A	41.36	34.97	23.67	/	/	42.04	31.42	26.54	/	/
B	18.10	18.21	8.76	51.22	3.71	40.66	18.52	18.57	20.27	1.98
C	17.26	8.06	13.67	56.93	4.08	20.24	10.25	15.62	50.25	3.64
D	20.44	9.09	11.23	55.26	3.98	22.68	23.15	11.25	40.05	2.87
E	43.08	18.60	38.32	/	/	41.26	19.16	39.58	/	/

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
