# Peer review of "Fabrication of Anti-Reflective Surface with Superhydrophobicity/High Oleophobicity and Enhanced Mechanical Durability via Nanosecond Laser Surface Texturing"

_materials, 2020, doi:10.3390/ma13245691_

Round 1

Reviewer 1 Report

The manuscript presents nanosecond laser-based surface texturing and chemical treatment on a steel alloy to impart anti-wetting and anti-reflective functionalities. It also presents a durability analysis of such functionalized surfaces. The manuscript reports some interesting results. However, the authors should address the following comments.

General comments:

  1. Either abbreviation (FOTS) should be defined or removed from the abstract.
  2. The introduction is too lengthy to follow. Some paragraphs can be divided into smaller ones. In particular, the authors should focus on relevant literature. When authors claim that the presented method is time-efficient and cost-effective, then they should focus on comparing it with the other methods. For example, why should one prefer laser surface texturing over chemical etching?
  3. When anti-reflective functionality is the focus of the paper, the authors should state a reason why this particular steel alloy is chosen. Is there any relevance to this material in real-world applications with regard to anti-reflective functionality?
  4. Why chemical treatment is needed when nanosecond laser texturing itself could lead to an increase in contact angles over a period of time. There is sufficient evidence in the literature where nanosecond laser surface texturing has shown to increase the contact angle with time. Please comment.

Specific comments:

  1. The schematic of the experimental set-up in Figure 1 is vague. What is the make of laser source? What is the scan head specification?
  2. What is the area of laser processing on substrates?
  3. How the overlap is evaluated? Can the authors point to a reference?
  4. What is the average fluence for grooves and dimples? Please mention
  5. Which instrument is used to evaluate the roughness? Contact type or non-contact type? Please mention.
  6. Is the roughness evaluated over a line or area? If it is over a line, then it is better to show the profile line on grooves and dimples. It is particularly important because the orientation of the profile line changes the Ra on grooves.
  7. Inset figures in Fig. 5 don’t make any sense! They could be removed.
  8. I think surface area fraction is a better parameter/key factor to differentiate grooves and dimples rather than the Ra. Can the authors comment?
  9. The refractive indices of the chosen steel alloy as well as air should be mentioned.
  10. What is the Ra of samples after the durability test? Is there any correlation between contact angle and Ra after each of the 10 abrasion cycles?

Author Response

The authors appreciate the reviewer’s valuable comments.

The edited part in the manuscript has been highlighted with yellow color.

Detailed response for each question can be seen on the table below below:

General Comments:

Comments:

Responses:

1 Either abbreviation (FOTS) should be defined or removed from the abstract.

The abbreviation FOTS was removed from the abstract in the revised manuscript.

2 The introduction is too lengthy to follow. Some paragraphs can be divided into smaller ones. In particular, the authors should focus on relevant literature. When authors claim that the presented method is time-efficient and cost-effective, then they should focus on comparing it with the other methods. For example, why should one prefer laser surface texturing over chemical etching?

Thanks for the reviewer’s suggestion. We have restructured the paragraphs in the introduction section and highlight the high time efficiency and low production cost of the laser texturing methods compared with the traditional coating and nanofabrication methods. The modified sections have been highlighted in yellow in the revised manuscript.

3 When anti-reflective functionality is the focus of the paper, the authors should state a reason why this particular steel alloy is chosen. Is there any relevance to this material in real-world applications with regard to anti-reflective functionality?

The anti-reflective steel alloy can be used for production of some photothermal devices such as thermoelectric generators, electro-optical cells and solar hot water tanks. The relevant content has been included in the introduction section.

4 Why chemical treatment is needed when nanosecond laser texturing itself could lead to an increase in contact angles over a period of time. There is sufficient evidence in the literature where nanosecond laser surface texturing has shown to increase the contact angle with time. Please comment.

Although storage in air can also lead to the increase of water contact angle over time, it can be a quite lengthy process that takes 10-30 days. The purpose of using chemical treatment in this work is to achieve fast transition of surface wettability for the laser textured surface, as the surface will become superhydrophilic immediately after laser texturing and superhydrophobic immediately after chemical treatment which takes about one hour. This will help to greatly improve the processing efficiency.

Specific Comments:

Comments:

Responses:

1 The schematic of the experimental set-up in Figure 1 is vague. What is the make of laser source? What is the scan head specification?

Both the laser source and scan head were manufactured by Han’s Laser and integrated into the laser marking machine. The schematic has been modified in Figure 1 as shown in the revised manuscript.

2 What is the area of laser processing on substrates?

The laser process area is 20 mm × 20 mm in this study, and it has been added in the revised manuscript.

3 How the overlap is evaluated? Can the authors point to a reference?

The overlap is calculated in the equations. However, due to the system error, the detailed equations and figures can be shown in the uploaded word file. Please see the attachment. 

4 What is the average fluence for grooves and dimples? Please mention.

The spacing between two grooves is 0.8 mm, thus the density (average fluence) for groove pattern is 12.5 line per cm.

Meanwhile, the spacing between spots in X and Y directions are also 0.8 mm, thus the density (average fluence) for spot pattern is 156 spots per cm2.

All these results have been included in the revised manuscript.

5 Which instrument is used to evaluate the roughness? Contact type or non-contact type? Please mention.

The surface roughness was measured by a non-contact 3-D laser scanning confocal microscope (OLS5000, Olympus, Japan), and the details have been included in section 2.2 in the revised manuscript.

6 Is the roughness evaluated over a line or area? If it is over a line, then it is better to show the profile line on grooves and dimples. It is particularly important because the orientation of the profile line changes the Ra on grooves.

The roughness was tested over a line, and the test direction was vertical to the groove/spot pattern. The explanation has been added in the revised manuscript. 

7 Inset figures in Fig. 5 don’t make any sense! They could be removed.

The real surface images of the surface in Figure 5 has been removed in the revised manuscript.   

8 I think surface area fraction is a better parameter/key factor to differentiate grooves and dimples rather than the Ra. Can the authors comment?

Thanks for the reviewer’s comments. We agree that the surface area fraction can be used to evaluate the surface roughness. However, in this study, due to the periodicity of the laser induced surface pattern, we believe the line roughness Ra can also be used to represent the surface roughness.

9 The refractive indices of the chosen steel alloy as well as air should be mentioned.

The refractive indices of 1095 steel and air are about 128 and 1. These details have been included in the revised manuscript in section 3.2.

10 What is the Ra of samples after the durability test? Is there any correlation between contact angle and Ra after each of the 10 abrasion cycles?

We did not test the Ra value after the abrasion test. However, we estimated that surface roughness has very minimal change before and after the durability test. This can be verified by the stability of water contact angle value during the durability test. As surface roughness is a key factor for achieving superhydrophobicity, it is believed that the surface roughness remained almost unchanged during the durability test. This also indicates the strong correlation between the contact angle and surface roughness for each 10 abrasion cycles.

Reviewer 2 Report

This is an interesting manuscript dedicated to laser surface texturing for modification of surface wettability. I have the following comments thereupon:

  1. Surface wettability modification is an interesting method for surfaces used in an atmosphere. The importance of this topic is beyond any doubt. For solar cells it is important that their protective coating be at the same time anti-reflective within a broad spectral range and non-wettable. These two primary requirements are sometimes broadened to also include abrasion resistance. This is achieved in the reviewed work by laser-chemical surface treatment. It is quite interesting, however, what the proposed method of protective surface treatment provides in comparison with other methods, what are its advantages. Could the anti-reflective surface by itself be non-wettable and abrasion-resistant? This question should be answered with adequate substantiation.
  2. Surface structuring with a laser beam is a lengthy process, particularly when the surface area is large. The reviewed work is concerned with the fundamental principles of such treatment. Nevertheless, from the practical point of view, the required duration of laser-chemical treatment is also quite important. This should be estimated and the proposed method compared with the conventional methods of surface treatment (for solar panels and so on).

3 The Conclusion mentions a relatively large number of abrasion cycles. However, such cycles may differ in the abradant grade, speed, &c. Therefore, average figures may not be applicable here. It is necessary to express surface abrasion resistance in different units.

If the revised version of the submitted manuscript takes into consideration the provided comments, it may be published in Materials.

Author Response

The authors appreciate the reviewer’s valuable comments.

The edited part in the manuscript has been highlighted with yellow color.

Detailed response for each question can be seen on the table below:

Comments:

Responses:

1 Surface wettability modification is an interesting method for surfaces used in an atmosphere. The importance of this topic is beyond any doubt. For solar cells it is important that their protective coating be at the same time anti-reflective within a broad spectral range and non-wettable. These two primary requirements are sometimes broadened to also include abrasion resistance. This is achieved in the reviewed work by laser-chemical surface treatment. It is quite interesting, however, what the proposed method of protective surface treatment provides in comparison with other methods, what are its advantages. Could the anti-reflective surface by itself be non-wettable and abrasion-resistant? This question should be answered with adequate substantiation.

Compared with other methods used for protective surface treatment, laser-based surface texturing methods have the following advantages:

1.     Easy for automation, highly reliable and repeatable process controlled by a modern computerized system;

2.     Can process difficult-to-reach spaces on final products;

3.     Increase the production precision;

4.     Lead to a high level of control over surface micro/nanostructures;

5.     Reduce waste generation, thus significantly reducing environmental impacts of surface texturing;

6.     Increase the surface durability and reliability, which can be a major issue for electrochemical and PVD coating methods as the coatings can easily get pilled off.

The anti-reflective property of surface is mainly determined by the specific surface micro/nanostructures which leads to the diffuse reflection and multiple specular reflection of light, thus resulting in the decrease of surface spectral reflectance [1]. However, it has been found that both surface micro/nanostructures and surface chemistry are equally essential to achieve a non-wettable surface. This means an anti-reflective surface is not necessarily non-wettable as well. Post-process treatment methods, such as chemical immersion, are required to change the surface chemistry of a micro/nanostructure surface, thus achieving superhydrophobicity.

The abrasion resistance is determined by both surface micro/nanostructure and surface hardness [2,3]. Therefore, traditional anti-reflective devices are not necessarily abrasion-resistant by itself. Therefore, surface modification methods are required to increase the mechanical strength of such surfaces.

2 Surface structuring with a laser beam is a lengthy process, particularly when the surface area is large. The reviewed work is concerned with the fundamental principles of such treatment. Nevertheless, from the practical point of view, the required duration of laser-chemical treatment is also quite important. This should be estimated and the proposed method compared with the conventional methods of surface treatment (for solar panels and so on).

Firstly, the duration of the laser-based fabrication method in this study is relatively short, comparing with the other ultrafast laser texturing methods, which has been illustrated in the manuscript.

Secondly, when fabricating a 100 mm × 100 mm surface, the duration of traditional surface coating or nanofabrication methods was about 3-5 hours [4,5], while the duration of the laser texturing process in this study was only 25 s for the groove pattern. Even with the subsequent chemical immersion process which will take effect in one hour, the whole laser-chemical treatment can be done in about an hour. This clearly indicates that the efficiency of the fabrication process in this study has been improved compared with the other conventional surface treatment methods.

3 The Conclusion mentions a relatively large number of abrasion cycles. However, such cycles may differ in the abradant grade, speed, &c. Therefore, average figures may not be applicable here. It is necessary to express surface abrasion resistance in different units.

Thanks for the reviewer’s valuable comments.

This test results cannot be evaluated by the exact value of abrasion resistance with a specific unit. We have reviewed a number of papers that conducted the similar work to evaluate the mechanical durability [6–9], and these relevant literatures show that the abrasion test is the proper method to test the surface durability (surface abrasion resistance), which can be both quantitively and qualitively evaluated using the key surface functionalities, such as surface wettability and spectral reflectance before and after the abrasion test.

In order to evaluate the change of mechanical durability during all the abrasion test, we had tested the water contact angle on the laser-chemical treated surface after every 10 abrasion cycles, while the results showed that the wettability almost do not change during 120 cycles.

At the same time, we have tried to test the reflectance of the laser-chemical surface during the abrasion test, while there was only small difference between 10 abrasion cycles. It can also be seen in Figure 7a that the reflectance of the laser-chemical treated superhydrophobic surface did not change a lot after 120 abrasion cycles. Thus, the authors believe that the reflectance after 120 cycles was clearly enough to illustrate the mechanical stability of the laser-chemical treated surface.

Reference:

  1. Vorobyev, A.Y.; Guo, C. Reflection of femtosecond laser light in multipulse ablation of metals. J. Appl. Phys. 2011, 110, 1–9, doi:10.1063/1.3620898.
  2. Buchely, M.F.; Gutierrez, J.C.; Le, L.M.; Toro, A. The effect of microstructure on abrasive wear of hardfacing alloys. Wear 2005, 259, 52–61, doi:10.1016/j.wear.2005.03.002.
  3. Nurminen, J.; Näkki, J.; Vuoristo, P. Microstructure and properties of hard and wear resistant MMC coatings deposited by laser cladding. Int. J. Refract. Met. Hard Mater. 2009, 27, 472–478, doi:10.1016/j.ijrmhm.2008.10.008.
  4. Li, Y.; Zhang, J.; Yang, B. Antireflective surfaces based on biomimetic nanopillared arrays. Nano Today 2010, 5, 117–127, doi:10.1016/j.nantod.2010.03.001.
  5. Copolymers, S.B.; Sahoo, P.K.; Tocce, E.; Auzelyte, V.; Ekinci, Y.; Solak, H.H.; Liu, C.; Stuen, K.O.; Nealey, P.F.; David, C. Nanofabrication of Broad-Band Antire fl ective Surfaces Using Self-Assembly of Block Copolymers Birgit. ACS Nano 2011, 1860–1864, doi:10.1021/nn103361d.
  6. Jannatun, N.; Taraqqi-A-Kamal, A.; Rehman, R.; Kuker, J.; Lahiri, S.K. A facile cross-linking approach to fabricate durable and self-healing superhydrophobic coatings of SiO2-PVA@PDMS on cotton textile. Eur. Polym. J. 2020, 134.
  7. Kusano, E.; Kitagawa, M.; Kuroda, Y.; Nanto, H.; Kinbara, A. Adhesion and hardness of compositionally gradient TiO2/Ti/TiN, ZrO2/Zr/ZrN, and TiO2/Ti/Zr/ZrN coatings. Thin Solid Films 1998, 334, 151–155, doi:10.1016/S0040-6090(98)01134-1.
  8. Lee, J.W.; Wang, H.C.; Li, J.L.; Lin, C.C. Tribological properties evaluation of AISI 1095 steel chromized at different temperatures. Surf. Coatings Technol. 2004, 188189, 550–555, doi:10.1016/j.surfcoat.2004.07.011.
  9. Lu, Z.; Wang, P.; Zhang, D. Super-hydrophobic film fabricated on aluminium surface as a barrier to atmospheric corrosion in a marine environment. Corros. Sci. 2015, 91, 287–296, doi:10.1016/j.corsci.2014.11.029.

Reviewer 3 Report

1. Introduction is too long and general (lines 37-66). Table 1 can be deleted (based on only seven citations). This data on the comparison of methods can be briefly mentioned in the text only.

2. The conditions of the scratch test are not clearly defined. The performed test corresponds more to the tribological test, where in addition to the load, the linear speed of the load displacement is also decisive. But the linear speed is not specified. Wear is indirectly compared to the change in reflectance and water contact angle.

3. The chemical composition of the surface before and after the wear test was not evaluated (e.g. XPS). It has a decisive influence on the change of the contact angle.

4. Surface nanostructure (structure below 100 nm) is not visible from the SEM images.

5. Roughness (Figure 3) given only in Ra without error bars.

6. Most of the same data has already been published in the cited article [32]. It might be worthwhile to better design and evaluate a surface wear experiment.

Author Response

The authors appreciate the reviewer’s valuable comments.

The edited part in the manuscript has been highlighted with yellow color.

Detailed response for each question can be seen in the table below:

General Comments:

Comments:

Responses:

1. Introduction is too long and general (lines 37-66). Table 1 can be deleted (based on only seven citations). This data on the comparison of methods can be briefly mentioned in the text only.

Thanks for the reviewer’s suggestions. We have restructured the paragraphs in the introduction section to improve the readability of the introduction section. Meanwhile, Table â…  has been deleted and we have provided a brief comparison for the different fabrication processes. The modified sections have been highlighted in yellow in the revised manuscript.

2. The conditions of the scratch test are not clearly defined. The performed test corresponds more to the tribological test, where in addition to the load, the linear speed of the load displacement is also decisive. But the linear speed is not specified. Wear is indirectly compared to the change in reflectance and water contact angle.

The linear speed of the scratch test has been defined in the revised manuscript.

3. The chemical composition of the surface before and after the wear test was not evaluated (e.g. XPS). It has a decisive influence on the change of the contact angle.

The chemical composition of the laser-chemical treated surfaces before and after the wear test have been evaluated by EDS. The results have been added in the revised manuscript.

4. Surface nanostructure (structure below 100 nm) is not visible from the SEM images.

SEM images with high magnification has been inserted as an inset figure in Figure 4 in the revised manuscript.

5. Roughness (Figure 3) given only in Ra without error bars.

We have removed the Ra values and added Sa values for all the surfaces. The error bar for all Sa values has been added in the revised manuscript in Figure 3.

6. Most of the same data has already been published in the cited article [32]. It might be worthwhile to better design and evaluate a surface wear experiment.

In paper [32], we proved that the developed laser-chemical treatment process is an effective method for fabrication of superhydrophobic surface. At the same time, we have shown that the fabricated surface exhibits excellent anti-corrosion properties, which have been proved using the results obtained by deliquescence test and electrochemical corrosion test.

In this paper, we aimed to discover more surface functionalities of the laser-chemical treated surface, including surface wettability of different liquids (water, glycerol, and engine oil), spectral reflectance and mechanical durability. The main results, discussions and conclusions are quite different for these two papers.

Meanwhile, the scratch test was designed based on a number of related literatures that conducted the similar work to evaluate the mechanical durability [1–4], and these relevant literatures show that the abrasion test is the proper method to test the surface durability (surface abrasion resistance), which can be both quantitively and qualitively evaluated using the key surface functionalities, such as surface wettability and spectral reflectance before and after the abrasion test.

Reference:

  1. Jannatun, N.; Taraqqi-A-Kamal, A.; Rehman, R.; Kuker, J.; Lahiri, S.K. A facile cross-linking approach to fabricate durable and self-healing superhydrophobic coatings of SiO2-PVA@PDMS on cotton textile. Eur. Polym. J. 2020, 134.
  2. Kusano, E.; Kitagawa, M.; Kuroda, Y.; Nanto, H.; Kinbara, A. Adhesion and hardness of compositionally gradient TiO2/Ti/TiN, ZrO2/Zr/ZrN, and TiO2/Ti/Zr/ZrN coatings. Thin Solid Films 1998, 334, 151–155, doi:10.1016/S0040-6090(98)01134-1.
  3. Lee, J.W.; Wang, H.C.; Li, J.L.; Lin, C.C. Tribological properties evaluation of AISI 1095 steel chromized at different temperatures. Surf. Coatings Technol. 2004, 188189, 550–555, doi:10.1016/j.surfcoat.2004.07.011.
  4. Lu, Z.; Wang, P.; Zhang, D. Super-hydrophobic film fabricated on aluminium surface as a barrier to atmospheric corrosion in a marine environment. Corros. Sci. 2015, 91, 287–296, doi:10.1016/j.corsci.2014.11.029.

Reviewer 4 Report

In my opinion, the paper entitled “Fabrication of Anti-reflective Surface with Superhydrophobicity/High Oleophobicity and Enhanced Mechanical Durability via Nanosecond Laser Surface Texturing”, has scientific interest and originality in its technical content to merit publication. The paper presents some errors that can be improved. I am sending a copy of the manuscript (pdf) in which all the suggestions/corrections are highlighted (green underscore).

Author Response

The authors appreciate the reviewer’s valuable comments.

The edited part in the manuscript has been highlighted with yellow color.

Detailed response for each question can be seen in the table below:

General Comments:

Comments:

Responses:

1. Line 68-69, rearrange the meaning of the sentence.

This sentence has been modified.

2. Line 122, since the authors decided to use the metric system rather than the British system of units, convert also the dimensions of the specimens to it.

All the dimensions of the specimens have been unified to the metric system.

3. Line 123, How are the tribological properties related with the present investigation?

The authors must comment on this!

The tribological properties of AISI 1995 steel are not directly related the surface functionalities of the laser-chemical treated surfaces in this study. Here, we just would like to show that 1095 steel alloy has many fascinating physical, chemical and mechanical properties, and this is why we used this material as the base material in our study.

4. Line 139, this information was already given.

It has been deleted in the revised manuscript.

5. Line 142, Figure 1a shows the Han´s Laser and not the reagent.

This error has been removed in the revised manuscript.

6. Line 157-159, It would be interesting to see the STDEV values since in my opinion we can not use the Ra roughness parameter to characterize textured surfaces as I explained in page 6.

Thanks for the reviewer’s suggestions.

We have double checked the surface roughness results. To better demonstrate the roughness results, the Ra values have been removed and Sa values have been added to evaluate the surface roughness of the samples.

7. Line 172-174, It would be interesting to have a table in order to compare the properties of different liquids: viscosity, density,...etc.

Thanks for the reviewer’s valuable comments.

A comparison table for the properties of water, glycerol and engine oil has been added in the revised manuscript.

8. Line 205-213, It is completely wrong to use Ra roughness parameter to compare smooth surfaces and textured surfaces. In fact, the chosen measurement conditions such as sampling length or cutoff lc filter may not even "see" the textures. 

The only way to characterize textured surfaces is using parameters like, dimple depth, dimple diameter, dimple density or groove depth and groove width, etc…or try to correlate the textured surfaces parameters with roughness parameters like Skewness and Kurtosis.

Thanks for the reviewer’s suggestions.

We have double checked the roughness results. To better demonstrate the roughness results, the Ra values have been removed and Sa values have been added to evaluate the surface roughness of the samples.

Round 2

Reviewer 3 Report

ad 1
Deleting of the table helped, but the introduction still contains redundant information.

ad 2
OK

ad3
The WDS method analyzes the chemical composition of the surface, including the volume to a depth of about 1 μm, and may be suitable for wear assessment. However, not for evaluating the contact angle. The contact angle strongly depends on the chemical composition of the surface. Why was the contact angle not measured after wear?

ad 4
OK

ad 5
Sa is more suitable for this type of surface, but in the case of surfaces treated with laser (sharp irregularities) and chemically etched (rounded edges), the information about roughness should be supplemented by the maximum height Sz. Sz is more sensitive to the surface wear.

ad 6
The wear evaluation is based on an indirect method. Probably the easiest is to analyze the surface before and after wear using SEM. The surface before wear was presented, but not after wear. It is not clear why the surface after wear was not presented. The contact angle was measured also only before the abrasive test.

Author Response

We appreciate the valuable comments from the reviewer.

We have highlighted the edited part in the revised manuscript with yellow color, and the detailed response can be seen in the table below:

Comments:

Responses:

ad 1
Deleting of the table helped, but the introduction still contains redundant information.

We have shortened the introduction section one more time. After this modification, the introduction only contains 939 words. We believe that the length of the introduction section in its current state is suitable for a research paper.

There are 5 paragraphs in the introduction section, and we believe all of the paragraphs possess information for this manuscript.

The outline for each paragraph can be found below:

·         Paragraph 1 provides background information for energy crisis.

·         Paragraph 2 introduces the importance of anti-reflective surface for solar energy collection and the existing issues with fabrication of anti-reflective surface.

·         Paragraph 3 provides a literature review for improvement of surface properties for anti-reflectance surface, which shows that anti-reflective surface with superhydrophobicity and enhanced mechanical durability can be utilized to solve the above mentioned problem.

·         Paragraph 4 discusses the fabrication techniques for multifunctional surface and shows that laser-based technique is promising, while technical improvement is still necessary to further enhance time and cost efficiency.

·         Paragraph 5 briefly introduces our developed laser-chemical treatment method, and also discusses our experimental results and the applications of our developed technique.

Overall, we believe that the introduction section has a very proper length at its current state.

ad 3
The WDS method analyzes the chemical composition of the surface, including the volume to a depth of about 1 μm, and may be suitable for wear assessment. However, not for evaluating the contact angle. The contact angle strongly depends on the chemical composition of the surface. Why was the contact angle not measured after wear?

The water contact angle values before and after wear test have been shown in our previous manuscript (Figure 7b). The water contact angle was recorded at every 10 scratch cycles until the completion of 120 cycles.

The chemical composition of the surface was evaluated to illustrate that the FOTS layer for sample B was damaged while the FOTS layer for sample C was more stable during the scratch test.

With the help of contact angle measurement and chemical composition measurement results, the mechanical stability of the laser-chemical treated samples can be verified clearly.

ad 5
Sa is more suitable for this type of surface, but in the case of surfaces treated with laser (sharp irregularities) and chemically etched (rounded edges), the information about roughness should be supplemented by the maximum height Sz. Sz is more sensitive to the surface wear.

Thanks for the reviewer’s comment.

The values of Sz have been added in Figure 3d in the revised manuscript.

Combining the values of Sa and Sz, the sharp irregularities from laser texturing and the rounded edges from chemical immersion can be specifically illustrated.

ad 6
The wear evaluation is based on an indirect method. Probably the easiest is to analyze the surface before and after wear using SEM. The surface before wear was presented, but not after wear. It is not clear why the surface after wear was not presented. The contact angle was measured also only before the abrasive test.

The SEM images of the laser-chemical treated surface before and after abrasion test have been added in the revised manuscript.

Meanwhile, the water contact angle values were recorded at every 10 scratch cycles until the completion of 120 cycles as shown in Figure 7b.
